# Attenuation of Inflammation and Leptin Resistance by Pyrogallol-Phloroglucinol-6,6-Bieckol on in the Brain of Obese Animal Models

**DOI:** 10.3390/nu11112773

**Published:** 2019-11-14

**Authors:** Myeongjoo Son, Seyeon Oh, Junwon Choi, Ji Tae Jang, Chang Hu Choi, Kook Yang Park, Kuk Hui Son, Kyunghee Byun

**Affiliations:** 1Department of Anatomy & Cell Biology, Gachon University College of Medicine, Incheon 21565, Korea; mjson@gachon.ac.kr (M.S.); choijw88@gc.gachon.ac.kr (J.C.); 2Functional Cellular Networks Laboratory, College of Medicine, Department of Medicine, Graduate School and Lee Gil Ya Cancer and Diabetes Institute, Gachon University, Incheon 21999, Korea; seyeon8965@gachon.ac.kr; 3Aqua Green Technology Co., Ltd., Smart Bldg., Jeju Science Park, Jeju 63309, Korea; whiteyasi@gmail.com; 4Department of Thoracic and Cardiovascular Surgery, Gachon University Gil Medical Center, Gachon University, Incheon 21565, Korea; cch624@gilhospital.com (C.H.C.); kkyypark@gilhospital.com (K.Y.P.)

**Keywords:** *Ecklonia cava*, phlorotannins, pyrogallol-phloroglucinol-6,6-bieckol, obese, leptin resistance

## Abstract

Obesity induces inflammation both in the adipose tissue and the brain. Activated macrophage infiltration, polarization of macrophages to a more inflammatory type (M1), and increased levels of pro-inflammatory cytokines are related to brain inflammation, which induces leptin resistance in the brain. Pyrogallol-phloroglucinol-6,6-bieckol (PPB), a compound from *Ecklonia cava*, has anti-inflammatory effects. In this study, we evaluated the effects of PPB effect M1 polarization and inflammation and its ability to restore the effects of leptin, such as a decrease in appetite and body weight. We administered PPB to diet-induced obesity (DIO) and leptin-deficient (ob/ob) mice, evaluated macrophage activation, polarization, and changes of inflammatory cytokine level in adipose tissue and brain, and determined the effect of PPB on leptin resistance or leptin sensitivity in the brain. The levels of activated macrophage marker, M1/M2, and pro-inflammatory cytokines were increased in the adipose tissue and brain of DIO and ob/ob mice than control. TLR4 expression, endoplasmic reticulum (ER) stress, and NF-κB expression in the brain of DIO and ob/ob mice were also increased; this increase was related to the upregulation of SOCS3 and decreased phosphorylated STAT3, which decreased leptin sensitivity in the brain. PPB decreased inflammation in the brain, restored leptin sensitivity, and decreased food intake and weight gain in both DIO and ob/ob mice.

## 1. Introduction

Leptin, the most well-known adipokine derived from adipocytes, regulates food intake and energy expenditure by binding to the long isoform of leptin receptor (Ob-R) in the central nervous system (CNS), mainly in the hypothalamic nuclei; it plays a crucial role in the maintenance of body weight [1,2]. In genetically-induced obese mice (ob/ob) which cannot synthesize leptin, treatment with recombinant leptin is highly effective at decreasing appetite and reversing obesity [3,4]. On the other hand, in diet-induced obese (DIO) mice or obese humans, leptin fails to prevent weight gain, even at high serum levels. Such unresponsiveness to endogenous or exogenous leptin is referred to as ‘leptin resistance’ [4,5]. Molecular leptin sensitivity, defined as the ability of leptin to phosphorylate and thereby activate STAT3 [3,6], is reduced after high-fat diet (HFD) intake [7]. The attenuation of the Janus-activating kinase2-signal transducer and activator of transcription 3 (JAK2-STAT3) signaling pathway is a crucial risk factor for leptin resistance [3,6,7]. 

The increase of suppressor of cytokine signaling-3 (SOCS3) expression is the primary inducer of leptin resistance in the CNS. SOCS3 potently inhibits the JAK/STAT pathway, thereby forming a negative feedback loop [8,9]. Obesity leads to chronic inflammation and changes the secretion patterns of adipokines, such as leptin in adipose tissue. In obese fat tissue, hypertrophic adipocytes exhibit increased expression and secretion of pro-inflammatory cytokines, including tumor necrosis factor α (TNF-α), interleukin (IL)-6, IL-8, and monocyte chemoattractant protein-1 [10,11]. Obesity leads to inflammation, not only in adipose tissue, but also in the CNS. The induction of pro-inflammatory TNF-α, IL-1β, and IL-6 mRNA, and protein expression in the hypothalamus during HFD feeding have been reported in animal models [12,13]. These pro-inflammatory cytokines inhibit leptin transport to the brain and lead to leptin resistance [14]. In addition, increased levels of SOCS3 mediate the hypothalamic leptin resistance induced by the activated inhibitor of nuclear factor kappa-B kinase subunit beta (IKKβ)/nuclear factor kappa-light-chain-enhancer of activated B cells (NF-κB) signaling [15]. Collectively, these data suggest that hypothalamic inflammation may contribute to obesity-associated leptin resistance in the brain. Emerging data demonstrate that, in addition to pro-inflammatory cytokines, microglial activation plays a pivotal role in the development of hypothalamic inflammation in obesity. Early after the introduction of an HFD, hypothalamic microglia undergo morphological and functional changes [16]. Microglial cells can sense changes in the environment and become activated to acquire either a pro-inflammatory (M1) or an anti-inflammatory (M2) phenotype [16,17]. M1 microglia cells are involved in increasing NF-kB signaling and secretion of pro-inflammatory cytokines such as TNF-α, IL-1β, and IL-6 [16].

*Ecklonia cava* (*E. cava*) is a type of marine brown alga which is known to be one of the richest natural sources of phlorotannins. Phlorotannins are a subclass of polyphenolic compounds that have a dibenzo-1,4-dioxin backbone, also dibenzodioxin, which makes the structure tight and promotes strong interactions with various biological molecules [18,19]. Some studies have shown that these compounds have various beneficial effects, including anti-inflammatory, antioxidative, and antidiabetic effects [20,21]. Pyrogallol-phloroglucinol-6,6-bieckol (PPB), a phlorotannin from E. cava, significantly inhibited monocyte migration in vitro by reducing the levels of inflammatory macrophage differentiation [18]. In the present study, we evaluated the attenuating effects of PPB on M1/M2 polarization and inflammation in the brain, which lead to a restoration of leptin actions such as decreasing appetite and controlling body weight by the modulation of leptin resistance or leptin sensitivity. We investigated macrophage activation and polarization, as well as changes in pro-inflammatory cytokines in the adipose tissue and brain of DIO mice and leptin-deficient (ob/ob) mice. We also determined the effect of PPB effect on leptin resistance in the brain.

## 2. Materials and Methods

### 2.1. Cell Culture

Murine Raw 264.7 macrophages were purchased from the Korean Cell Line Bank (Seoul, Korea). Murine 3T3-L1 preadipocytes were purchased from the American Type Culture Collection (Manassas, VA, USA). All cells were cultured in Dulbecco’s Modified Eagle Medium (DMEM; cat. LM001-07, Welgene, Gyeongsan, Korea) supplemented with 10% fetal bovine serum (Gibco; Grand Island, NY, USA) and 1% penicillin-streptomycin (Gibco) in a 5% CO2 incubator (Thermo Fisher Scientific, Waltham, MA, USA) at 37 °C; the medium was changed every two days.

### 2.2. Preparation of Palmitic Acid–Conjugated Bovine Serum Albumin and Cell Treatment

To prepare palmitic acid–conjugated bovine serum albumin (PA-BSA), 2.267 g of fatty acid-free BSA (BSA, cat. A8806; Sigma-Aldrich, St Louis, MO, USA) was dissolved in pre-warmed 100 mL of 150 mM sodium chloride. The mixture was stirred at 37 °C in a water bath until the BSA was completely dissolved, and the BSA solution was filtered into a new flask. While the BSA was being stirred, 30.6 mg of sodium palmitate (cat. P9767; Sigma-Aldrich) was dissolved in 50 mL of 150 mM sodium chloride in a water bath at 70 °C. The PA-BSA was divided into 5-mL aliquots and stirred at 37 °C for 1 h; the final volume was adjusted to 100 mL with 150 mM sodium chloride and the pH to 7.4 with 1 M sodium hydroxide. The solution was stored at −80 °C until required, and thawed in a 37 °C water bath for 5 min before use. Finally, 0.25 mM PA-BSA, with or without phlorotannins, were treated in Raw 264.7 cells and Murine 3T3-L1 cells.

### 2.3. Animals

#### 2.3.1. High-Fat-Diet-Induced Obese Mice

Seven-week-old male C57BL/6N mice were obtained from Orient Bio (Seongnam, Korea). The mice were housed under a dark–light cycle (12:12 h) at room temperature (23 °C) as described below. Drinking water was provided ad libitum for four weeks, and the mice were divided into three groups. The mice were fed either a normal fat diet (NFD; LabDiet, St Louis, MO, USA: 62% carbohydrates, 20% protein, and 5% fat, in % Kcal), or a high-fat diet (HFD; Research Diet Inc., New Jersey, NJ, USA: 35% carbohydrates, 20% protein, and 45% fat, in % Kcal) adapted from a previous study [22] for four weeks. Normal saline (0.9%; NFD-Saline and HFD-Saline groups) or 2.5 mg/kg PPB in normal saline (HFD-PPB group) was orally administered daily for four weeks, and normal saline was administered with the same volume of PPB by oral administration (Appendix A). After eight weeks, all mice were sacrificed in accordance with the ethical principles in the Institutional Animal Care and Use Committee of Gachon University (IACUC; approval number, LCDI-2017-0034).

#### 2.3.2. Leptin-Deficient Mice

Six-week-old male leptin-deficient obese (ob/ob) mice were obtained from Orient Bio, and were kept at a constant temperature around 23 °C and relative humidity of 50% under a dark–light cycle (12:12 h). Mice were fed a regular chow diet and drinking water ad libitum for seven weeks; one week after arrival, the mice were divided into four groups. To validate the effect of leptin or PPB, mice were intraperitoneally injected with 0.425 mg/kg of leptin (cat. 450-31; PeproTech, Rocky Hill, NJ, USA) twice a day (total 0.85 mg/kg/day), or PPB was orally administered twice a day (total 2.5 mg/kg/day) (Appendix A). This study was approved by the Lee Gil Ya Cancer and Diabetes Institute of Gachon University and was conducted in strict accordance with the guidelines issued by the IACUC of Gachon University (approval number: LCDI-2016-090).

Body weight, fat, and the lean body masses of all mice were measured with a minispec mice analyzer (cat. LF90II; Bruker Optik, GmbH, Germany).

### 2.4. Isolation of Compounds from E. cava Extract

To prepare the extract, *E. cava* collected near the Jeju coast was thoroughly washed with fresh water, air dried at room temperature for 48 h, finely ground, and extracted with 50% ethanol at 85 °C for 12 h. The extract was filtered, concentrated and sterilized at 85 °C for 40–60 min. Finally, dry powder was produced by spray-drying. Dieckol, 2,7-phloroglucinol-6,6-bieckol (PHB), phlorofucofuroeckol A (PFFA), and PPB, which are active compounds of *E. cava*, were isolated as previously described [23]. Briefly, centrifugal partition chromatography (CPC) was performed using a two-phase solvent system comprising ethyl acetate/water/methanol/n-hexane (7:7:3:2, *v/v/v/v*). A CPC column was first filled with organic stationary phase, and the mobile phase was pumped into the column in descending mode at the same flow rate (2 mL/min).

### 2.5. Quantitative Real-Time Polymerase Chain Reaction (qRT-PCR)

Total RNA was isolated using TRIzol (cat. 12183555; Invitrogen, Carlsbad, CA, USA) according to the manufacturer’s instructions. Tissue was homogenized on ice using a disposable pestle in 1 mL of TRIzol, 0.2 mL of chloroform (cat. 0757; Amresco, Solon, OH, USA) was added, mixed, and centrifuged at 12,000× *g* for 15 min at 4 °C. The aqueous phases were collected, placed in a clean tube, mixed with 0.5 mL of isopropanol, and centrifuged under the same conditions. Isolated RNA was then washed with 70% ethanol and dissolved in 30 μl of diethylpyrocarbonate (DEPC)-treated water. Total RNA (1 μg) was used for complementary DNA (cDNA) synthesis using the PrimeScript 1st Strand cDNA Synthesis Kit (cat. 6110A; Takara, Otsu, Japan). qRT-PCR was performed in a C1000 Touch thermal cycler (Bio-Rad, Hercules, CA, USA). All primers were designed using mouse-specific sequences, and are listed in Appendix A.

### 2.6. Immunoblotting

To extract protein, collected tissues were lysed using an EzRIPA lysis kit (cat. WSE-7420; ATTO, Tokyo, Japan), homogenized, and centrifuged at 13,000× *g* for 20 min at 4 °C. Supernatants were transferred to fresh tubes, and protein contents were determined using a bicinchoninic acid assay kit (cat. 23225; Thermo Fisher Scientific). Proteins (20 µg) were separated by 10% sodium dodecyl sulfate-polyacrylamide gel electrophoresis and transferred to polyvinylidene fluoride membranes using a Semi-Dry transfer system (ATTO) at 25 V for 10 min. The membranes were then blocked with 5% (*w/v*) skimmed milk in Tris-buffered saline (pH 7.6) containing 0.1% Tween 20 (TBST) for 2 h. The membranes were washed with TBST, incubated with primary antibodies in blocking solution overnight at 4 °C, washed with TBST twice, incubated with appropriate secondary antibodies, and rewashed. Proteins of interest were detected using EzWestLumi plus luminol substrate (ATTO) on an ImageQuant LAS-4000 imager (GE Healthcare, Uppsala, Sweden). The antibodies used are listed in Appendix A.

### 2.7. Fat size Analysis

Visceral (epididymal and perirenal) fat was collected. The fat tissues were fixed in 4% paraformaldehyde (cat. P2031; Biosesang, Seongnam, Korea) overnight at 4 °C and placed in an automatic dehydration machine (cat. ASP300S; Leica, Milton Keynes, UK). Tissues were dehydrated in 70% ethanol for 30 min, 80% ethanol for 30 min, 90% ethanol three times for 1 h, and 100% ethanol twice for 2 h. Tissues were then cleared with 100% xylene three times for 1.5 h each, and then embedded in warmed paraffin; 7-µm sections were cut and stained with hematoxylin and eosin (H&E). Tissues were deparaffinized in 100% xylene twice for 5 min, 100% ethanol twice for 3 min, 90% ethanol twice for 3 min, 80% ethanol for 1 min, and 70% ethanol for 1 min. Tissues were washed with distilled water and then stained with hematoxylin (cat. S3309; DAKO, Tokyo, Japan) for 1 min and eosin (cat. 318906; Sigma-Aldrich) for 30 sec. Images were taken using an Axio Imager Z1 upright microscope, and adipocyte size was quantified using ImageJ 1.50i software (NIH, Bethesda, MD, USA).

### 2.8. Statistical Analysis

A statistical analysis was conducted using SPSS version 22 software (IBM Co.; New York, NY, USA). In this study, statistical differences were compared among palmitatic acid-treated cell group (PBS, PA/PBS, PA/dieckol (DK), PA/PHB, PA/PPB, PA/PFFA), diet-induced obese groups (NFD-Saline, HFD-Saline, HFD-PPB), and leptin-deficient groups (wild type (WT)-Saline, ob/ob-Saline, ob/ob-PPB, ob/ob-Leptin) using the non-parametric Kruskal-Wallis test. For the post-test, the differences between the two groups were compared using the Mann-Whitney U test. All results are presented as means ± standard deviation, and all experiments were repeated three times* 

* indicates comparing with BSA, NFD-Saline or WT-Saline.

$ indicates comparing with PA-BSA, HFD-Saline or ob/ob-Saline.

# indicates comparing with PA-PPB or ob/ob-PPB.

## 3. Results

### 3.1. PPB Decreases M1 Polarization and Production of Inflammatory Cytokines in Raw 264.7 Cells, and Decreases Adipogenesis and Lipogenesis in 3t3l-1 Cells More Efficiently than Other Components of E. cava Extracts

To choose the components of the *E. cava* extract with the greatest potential for decreasing inflammation, adipogenesis, or lipogenesis in adipocytes, we treated Raw 264.7 and 3T3L-1 cells with four different phlorotannins from the *E. cava* extracts. In Raw 264.7 cells, PPB was most efficient at decreasing the expression of CD11b (an activated-macrophage marker) [24], CD86 (a marker of M1 macrophages), TNF-α, and IL-6, and at increasing the expression of CD206 (a marker of M2 macrophages) (Figure 1A–E). The mRNA levels of an adipogenesis-related gene (for peroxisome proliferator-activated receptor gamma, PPARγ) and lipogenesis-related genes (for acetyl-CoA carboxylase, ACC; and fatty acid synthase, FAS) decreased more in the PPB-treated 3T3L-1 cells than in those treated with other phlorotannins (Figure 1F–H). Thus, we chose PPB for the evaluation of the effects of E. cava extracts on inflammation in the brain and weight loss.

### 3.2. PPB Reduces Activated Macrophage Infiltration, M1 Polarization, and Inflammatory Cytokine Expression Levels in the Adipose Tissue and Brain of High Fat Diet–Induced Obese Mice

In the visceral fat tissue of the HFD-Saline group, the expression of CD11b was higher than in the NFD-Saline group; PPB significantly attenuated CD11b expression (Figure 2A). CD86 expression was increased by HFD, and was significantly decreased by PPB (Figure 2A). CD206 expression was decreased by HFD and was significantly increased by PPB. TNF-α and IL-6 expression in visceral fat tissue were increased by HFD and decreased by PPB (Figure 2B). In addition, CD11c, as a well-known adipose tissue macrophages, was also validated. CD11c expression was increased by HFD, and was significantly decreased by PPB in visceral fat (Appendix A). In the brain of the HFD group, the expression of CD11b and CD86 was higher than in the brain of the NFD-Saline group, but was decreased in the HFD-PPB group (Figure 2C). In the brain, CD206 expression was decreased by HFD and increased by PPB, whereas TNF-α and IL-6 expression was increased by HFD and decreased by PPB (Figure 2D).

### 3.3. PPB Reduces Activated Macrophage Infiltration, M1 Polarization, and Inflammatory Cytokine Transcript Levels in the Adipose Tissue and Brain of Ob/Ob Mice

In visceral fat tissue, the expression of CD11 and CD86 was higher in ob/ob-Saline mice than in WT-Saline mice, and was decreased by the administration of leptin or, to a lesser extent, of PPB (see Figure 3A). In the visceral fat tissue, CD206 expression was lower in ob/ob-Saline mice than in WT-Saline mice, and was increased by PPB or leptin administration (Figure 3A), whereas TNF-α and IL-6 expression was increased in ob/ob mice but was decreased by PPB or leptin (Figure 3B). In addition, CD11c also validated. The CD11c expression was increased by ob/ob, and was significantly decreased by PPB in visceral fat (Appendix A). In the brain, the expression of CD11b and CD86 was higher in ob/ob than in WT mice, and was decreased by PPB or leptin administration, whereas the expression of CD206 was lower in ob/ob than in WT mice, and was increased by PPB or leptin administration (see Figure 3C). TNF-α and IL-6 expression in the brain was higher in ob/ob than in WT mice and was decreased by PPB or leptin (Figure 3D).

### 3.4. PPB Attenuates TLR4 Expression and Endoplasmic Reticulum (ER) Stress in the Brain of DIO and Ob/Ob Mice

The expression level of TLR4 in the brain of C57BL/6N mice was increased by HFD and was attenuated by PPB (Figure 4A). The expression of the ER stress markers PKR-like ER protein kinase (PERK), eukaryotic initiation factor 2 alpha (eIF2α), inositol-requiring protein 1 (IRE1), and X-box–binding protein 1 (Xbp1) in the brain was increased by HFD and was attenuated by PPB (Figure 4B–E). The expression level of TLR4 in the brain was higher in ob/ob than in WT mice and was decreased by leptin or PPB administration (Figure 4F). The levels of ER-stress markers in the brain were higher in ob/ob mice than in WT mice and were decreased by PPB or leptin administration in ob/ob mice (Figure 4G–J).

### 3.5. PPB Decreases the NF-κB Level in the Brain of DIO and Ob/Ob Mice

The NF-κB level in the brain of C57BL/6N mice was increased by HFD but was decreased by PPB (Figure 5A). The NF-kB level in the brain was higher in ob/ob than in WT mice and decreased by PPB or leptin administration in ob/ob mice (Figure 5B). 

### 3.6. PPB Attenuates Leptin Resistance in the Brain of DIO Mice

The SOCS3 level in the brain was increased by HFD but was decreased by PPB in DIO mice (Figure 5C). The pSTAT3 level in the brain was decreased by HFD, but was increased by PPB (Figure 5C). The Ob-R expression in the brain was decreased by HFD but was increased by PPB (Figure 5E).

### 3.7. PPB Increases Leptin Sensitivity in the Brain of Ob/Ob Mice

The SOCS3 level in the brain was higher in ob/ob-Saline than in WT-Saline mice, and was decreased by PPB or leptin administration (Figure 5D). The brain pSTAT3 level was lower in ob/ob than in WT mice and was decreased in ob/ob mice by administration of leptin or PPB. Ob-R expression was lower in ob/ob-Saline than in WT-Saline mice, and was increased by PPB or leptin administration (Figure 5F).

### 3.8. PPB Decreases Adipogenesis and Lipogenesis in the Adipose Tissue of DIO and Ob/Ob Mice

Expression of adipogenesis-related genes (for PPARγ and CEBP) and lipogenesis-related genes (for ACC and FAS) in visceral fat was increased by HFD but was decreased by PPB (Figure 6A,B). The levels of PPARγ, CEBP, ACC, and FAS mRNA in visceral fat were higher in ob/ob than in WT mice and were decreased by leptin or PPB administration in ob/ob mice (Figure 6C,D).

### 3.9. PPB Decreases Body Weight Gain, Fat Mass, Food Intake, and Visceral Fat Size

The body weight and fat mass were increased by HFD and were decreased by PPB (Figure 7A). The food intake amount was increased by HFD but was decreased by PPB (Figure 7B). The visceral fat size in visceral fat tissue was increased by HFD but was decreased by PPB (Figure 7C). Body weight, fat mass, and food intake were higher in ob/ob than in WT mice, and were decreased in ob/ob mice by administration of PPB or leptin (Figure 7D,E). Fat size in visceral fat tissue was larger in ob/ob-Saline mice than in WT-Saline mice, and was smaller in ob/ob-Saline mice administered leptin or PPB than in ob/ob-Saline mice (Figure 7F).

## 4. Discussion

In obese adipose tissue, adipocytes secrete pro-inflammatory cytokines [10,11], which is in line with our results that TNF-α and IL-6 expression in adipose tissue was increased by HFD but decreased by PPB. Low-grade inflammation in obesity also occurs in the CNS; even a short-term hypercaloric challenge, especially an HFD, can induce inflammation in the hypothalamus of DIO animals [9,25]. In rats, HFD feeding for 4 months activates hypothalamic inflammatory pathways, such as JNK and NF-κB, which leads to the production of pro-inflammatory cytokines such as IL-1β, TNF-α, and IL-6, and even to deficiencies in leptin signaling. These observations were confirmed in many animal studies [15,16,26]. In our study, TNF-α and IL-6 expression in the brain was increased by HFD but was decreased by PPB. HFD, during a much shorter period (24–72 h), is enough to induce gliosis (the activation and proliferation of glial cells such as microglia and astrocytes), an inflammation hallmark in the brain [9]. In our study, activated macrophage infiltration in the brain, which was confirmed by measuring the CD11b marker, was increased in DIO mice but was decreased by PPB. However, it is hard to tell how much the contributions to the signals are due to local macrophages and how much they are due to local adipocytes or brain cells. However, the purpose of this study was to verify that the oral administration of PPB indirectly regulates the inflammation of brain tissue by regulating the expression of inflammatory macrophages and inflammatory cytokines in visceral adipose tissue. Therefore, further study on the contributions to the signals from local macrophages and local adipocytes/brain cells is needed.

In the hypothalamus, TLR4 is predominantly expressed by microglia [27]. During chronic HFD feeding, hypothalamic TLR4 expression and activity are increased [13,28]. The TLR4 signaling pathway promotes microglia activation, resulting in switching to an M1-like pro-inflammatory phenotype, which secretes TNF-α and IL-6 [16]. In our study, M1 macrophage marker expression was increased in the brain of DIO mice, and was decreased by PPB. TLR is also related to ER stress. TLR4 signaling is crucial to the development of ER stress [29]. An ER stress inhibitor and a TLR4 blocker decrease the expression of TLR4, its downstream factors such as TRAF6, IKKβ, and inflammatory factors TNF-α and IL-6, indicating that TLR4 is the key signaling element of ER stress, regulating inflammation in adipose tissue [29,30]. IKKβ/NF-κB and ER stress promote each other during HFD intake, induce leptin resistance by up-regulating SOCS3, and promote the energy imbalance underlying obesity [15]. In our study, the expression of TLR4 and ER stress markers (PERK, eIF2α, IRE1, and Xbp1) was increased in the brain of DIO mice and, interestingly, was decreased by PPB. In addition to increasing TLR4 expression and ER stress, TNF-α promotes the onset of leptin resistance by further activation of the JNK/NF-κB signal transduction pathways. CNS exposure to low-dose TNF-α promotes leptin resistance [31]. Both TNF-α and NF-κB induce leptin resistance by up-regulating SOCS3 expression. In our study, the expression NF-κB and SOCS3 in the brain was increased in DIO mice but was decreased by PPB.

The phosphorylation of STAT3 induced by leptin is reduced after HFD intake [3]. In our study, STAT3 phosphorylation in the brain of DIO mice was decreased, but it was increased by PPB. An increase in Ob-R expression and its localization at the cell surface are key determinants of cell sensitivity to leptin [32]. In our study, the Ob-R expression was decreased in the brain of DIO mice but was increased by PPB.

Our results suggest that PPB might decrease the levels of pro-inflammatory cytokines, infiltration of activated macrophages, and M1/M2 polarization in both the brain and adipose tissue. PPB also decreased TLR4 expression, ER stress, and NF-κB expression in the brain, all of which were increased by HFD. Therefore, PPB decreased inflammation in the brain and restored leptin sensitivity. The restoration of leptin sensitivity, which was confirmed by the decrease in the SOCS3 level and increase in pSTAT3 and Ob-R levels, decreased food intake, weight gain, and body fat mass increased by HFD. PPB also decreased adipogenesis and lipogenesis in visceral fat tissue. Studies evaluating adipose tissue or hypothalamus inflammation in genetically-obese mice are rare in comparison with studies focused on the inflammatory effect by HFD. Santoro et al. showed that ob/ob mice exhibit less microglial activation than WT controls, both on normal chow and on HFD [32]. The lack of leptin signaling affects microglial function in the hypothalamus, as the expression of several inflammatory mediators is reduced [33]. However, another study showed that leptin prevents the obesity-associated inflammatory state and the increased oxidative stress in leptin-deficient ob/ob mice [34]. Adipose tissue from ob/ob mice exhibited a dramatic increase in the expression of genes involved in inflammation in comparison with WT mice [34]. Leptin administration significantly reduces the expression of IL-6 or TNF-α, in ob/ob mice in comparison with vehicle-treated littermates [34].

In our study, the expression of pro-inflammatory cytokines (TNF-α, IL-6) was increased in the adipose tissue of ob/ob mice but was decreased by PPB, even though the effect of PPB was weaker than that of leptin. The levels of TNF-α and IL-6 mRNA in the brain of were also higher in ob/ob than in WT mice and were decreased by PPB in ob/ob mice. Inflammatory responses mediated by TNF-α signaling in the hypothalamus are integrally involved in obesity in genetical leptin–deficient obese models [35]. Both IL-6 and TNF-α levels can be increased in obesity [36,37], and both cytokines can cross the blood–brain barrier by a saturable transport mechanism [38,39]. This means that TNF-α or IL-6 secreted from adipose tissue or hypothalamus could induce or aggravate the inflammation in the hypothalamus of ob/ob mice. In our study, activated macrophage infiltration and M1/M2 polarization were increased in ob/ob mice and were decreased by PPB, although to a lesser extent than by leptin. As in the DIO model, the expression of TLR4 and NF-kB, and ER stress were increased in ob/ob mice and were decreased by PPB, although to a lesser extent than by leptin. Increased TLR4, ER stress, and NF-kB could induce inflammation, which would increase SOCS3 in ob/ob mice. SOCS3 expression was increased in ob/ob mice but was decreased by PPB, although to a lesser extent than by leptin. Ob-R expression was decreased in ob/ob mice but was increased by PPB, although to a lesser than by leptin. The increase in leptin sensitivity induced by PPB decreased food intake, weight loss, and fat mass in ob/ob mice. Although PPB seems to work in both brain and fat, the fact that PPB does not directly work in the brain seems to be the limitation of this study, and it is necessary to conduct similar studies in the future. Our results show that DK [40], one of the phlorotannins extracted from *E. cava* extract, passes through brain-blood barrier to brain parenchyma via unknown mechanisms, even though it has a number of polar groups and molecular weights over 700.

## 5. Conclusions

Our study showed that PPB decreases inflammation in the adipose tissue and brain; this effect is determined by a decrease in activated macrophage and M1 macrophage infiltration and an increase in TNF-α and IL-6 in both the DIO and ob/ob models. The pro-inflammatory cytokines in the brain increased TLR4 expression, ER stress, and NF-κB expression, which enhanced inflammation and induced leptin resistance. PPB restored leptin action in the brain by decreasing leptin resistance or increasing leptin sensitivity in both the DIO and ob/ob models, and thus, decreased food intake and weight gain.

## Figures and Tables

**Figure 1 nutrients-11-02773-f001:**
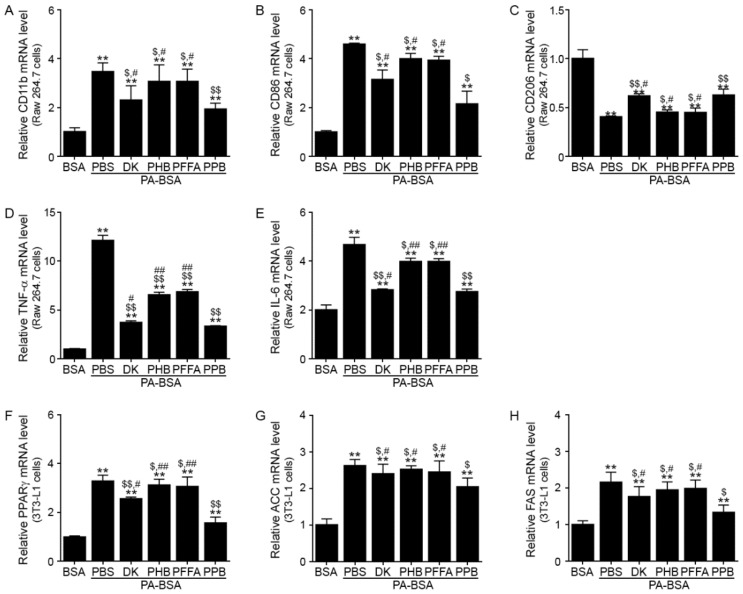
Reduction of M1 polarization and production of pro-inflammatory cytokines in Raw 264.7 cells, and of adipogensis/lipogenesis in 3T3L-1 cells by PPB from E. cava mRNA levels of (**A**) CD11b as a general macrophage marker, (**B**) CD86 as a marker of M1 macrophages, (**C**) CD206 as a marker of M2 macrophages, (**D**) TNF-α, and (**E**) IL-6 in Raw 264.7 cells were measured by qRT-PCR. Cells were pre-treated PA-BSA (0.25 mM) and four phlorotannins (DK, PHB, PFFA, and PPB) for 48 h. mRNA levels of (**F**) PPARγ, (**G**) ACC (adipogenesis-related markers), and (**H**) FAS (lipogenesis-related marker) in PA-BSA (0.25 mM) treated 3T3L-1 cells were measured by qRT-PCR. All mRNA levels are expressed as relative levels and are normalized to β-actin in the BSA group. Significance represented as **, *p* < 0.01 versus BSA; $, *p* < 0.05 and $$, *p* < 0.01 versus PA-BSA; #, *p* < 0.05 and ##, *p* < 0.01 versus PA-PPB. DK, dieckol; PHB, 2,7-phloroglucinol-6,6-bieckol, PFFA, phlorofucofuroeckol A; PPB, pyrogallol-phloroglucinol-6,6-bieckol; PA-BSA, palmitic acid–conjugated bovine serum albumin.

**Figure 2 nutrients-11-02773-f002:**
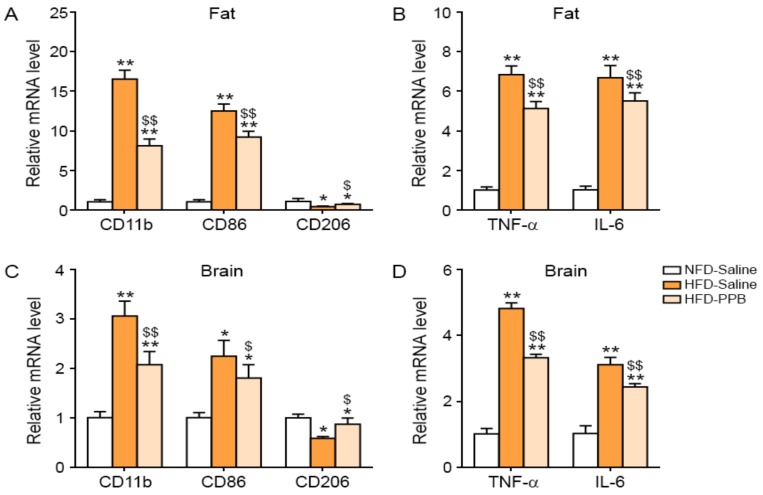
Reduction of activated macrophage infiltration, M1 polarization, and pro-inflammatory cytokine production, in the visceral fat and brain of the high fat diet–induced obese mice by PPB from *E. cava*. Three groups of diet-induced obese (DIO) mice were examined: normal fat diet with saline administration (NFD-Saline, white color), a high fat diet with saline administration (HFD-Saline, orange color), and HFD with PPB (2.5 mg/kg/day) administration (HFD-PPB). Saline and PPB were administered orally. mRNA levels of (**A**) CD11b as a general macrophage marker, CD86 as a marker of M1 macrophages, and CD206 as a marker of M2 macrophages, and (**B**) TNF-α and IL-6 in visceral fat tissue were measured by qRT-PCR. mRNA levels of (**C**) CD11b, CD86, and CD206 as a marker of M2 macrophages, and (**D**) TNF-α and IL-6 in brain were measured by qRT-PCR. All mRNA levels are expressed as relative levels normalized to β-actin of the NFD-Saline group. Significance represented as *, *p* < 0.05 and **, *p* < 0.01 versus NFD-Saline; $, *p* < 0.05 and $$, *p* < 0.01 versus HFD-Saline. PPB, pyrogallol-phloroglucinol-6,6-bieckol.

**Figure 3 nutrients-11-02773-f003:**
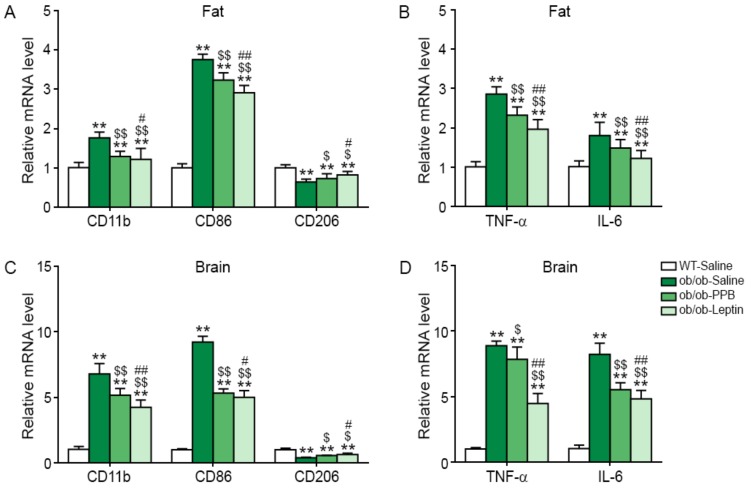
Reduction of activated macrophage infiltration, M1 polarization, and pro-inflammatory cytokine production in the visceral fat and brain of the ob/ob mice by PPB from *E. cava.* Three groups of leptin-deficient obese (ob/ob) mice and wild type (WT) control C57BL/6N mice were examined: Saline orally administered to WT (WT-Saline, white color) or ob/ob mice (ob/ob-Saline, green color). PPB (2.5 mg/kg/day, i.o., ob/ob-PPB), or leptin (0.85 mg/kg/day, i.p., ob/ob-Leptin) was administered to ob/ob mice. mRNA levels of (**A**) CD11b (general macrophage marker), CD86 (a marker of M1 macrophages), and CD206 (a marker of M2 macrophages) in visceral fat tissue and (**B**) TNF-α and IL-6 in the visceral fat tissue of were measured by qRT-PCR. mRNA levels of (**C**) CD11b, CD86, and CD206 in brain and (**D**) TNF-α and IL-6 in brain of were measured by qRT-PCR. All mRNA levels are expressed as relative levels normalized to β-actin of the WT-Saline group. Significance represented as **, *p* < 0.01 versus WT-Saline; $, *p* < 0.05 and $$, *p* < 0.01 versus ob/ob-Saline; #, *p* < 0.05 and ##, *p* < 0.01 versus ob/ob-PPB. PPB, pyrogallol-phloroglucinol-6,6-bieckol.

**Figure 4 nutrients-11-02773-f004:**
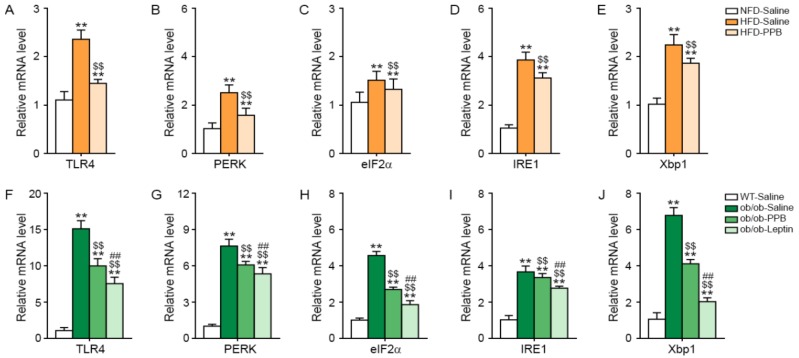
Reduction in TLR4 expression and endoplasmic reticulum (ER) stress in the brain of high fat diet–induced obese mice and ob/ob mice by PPB from *E. cava* mRNA levels of (**A**) TLR4 and the ER stress–related molecules including (**B**) PERK, (**C**) eIF2α, (**D**) IRE1, and (**E**) Xbp1 in the brain of high fat diet–induced obese mice (**A–E**) and (**F**) ob/ob mice were measured by qRT-PCR. mRNA levels of (**F**) TLR4 and the ER stress–related molecules including (**G**) PERK, (**H**) eIF2α, (**I**) IRE1, and (**J**) Xbp1 in the brain of (**F–J**) ob/ob mice were measured by qRT-PCR. All mRNA levels were expressed as relative levels normalized to β-actin of the NFD-Saline for high fat diet-induced obese mice group or WT-Saline for ob/ob mice. Significance represented as **, *p* < 0.01 versus NFD-Saline or WT-Saline; $$, *p* < 0.01 versus HFD-Saline or ob/ob-Saline; ##, *p* < 0.01 versus ob/ob-PPB. TLR4, Toll-like receptor 4; PERK, protein kinase R (PKR)-like ER protein kinase; eIF2α, eukaryotic initiation factor 2 alpha; IRE1, inositol-requiring protein 1 alpha; Xbp1, X-box–binding protein 1; PPB, pyrogallol-phloroglucinol-6,6-bieckol.

**Figure 5 nutrients-11-02773-f005:**
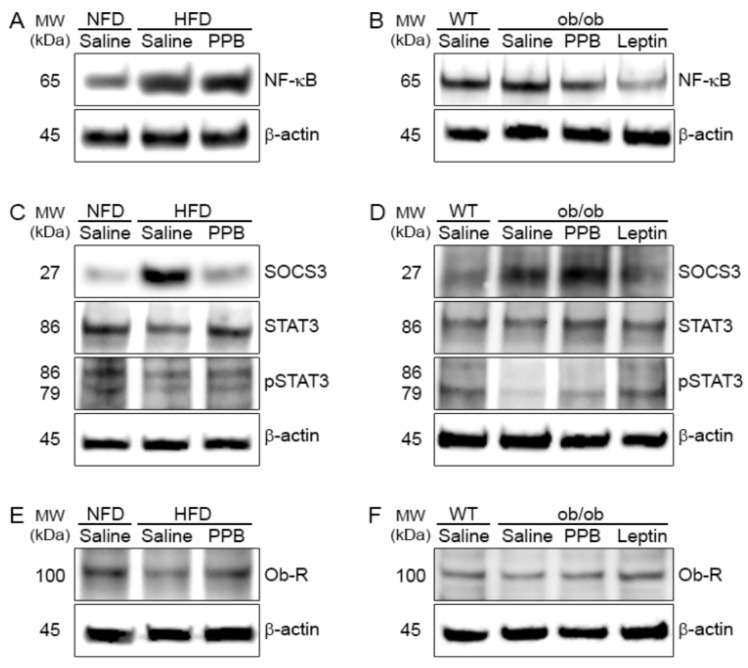
Modulation of NF-κB, SOCS3, pSTAT3, and Ob-R protein levels in the brain of high fat diet–induced obese mice and ob/ob mice by PPB *from E. cava* (**A**) NF-κB, (**C**) SOCS3, STAT3, and phosphorylated STAT3 (pSTAT3) and (**E**) Ob-R in the brain of high fat diet-induced obese. (**B**) NF-κB, (**D**) SOCS3, STAT3, and phosphorylated STAT3 (pSTAT3) and (**F**) Ob-R in the brain of ob/ob mice. Each protein level was determined by immunoblotting. NF-κB, nuclear factor-kappa B; SOCS3, suppressor of cytokine signaling 3; PPB, pyrogallol-phloroglucinol-6,6-bieckol.

**Figure 6 nutrients-11-02773-f006:**
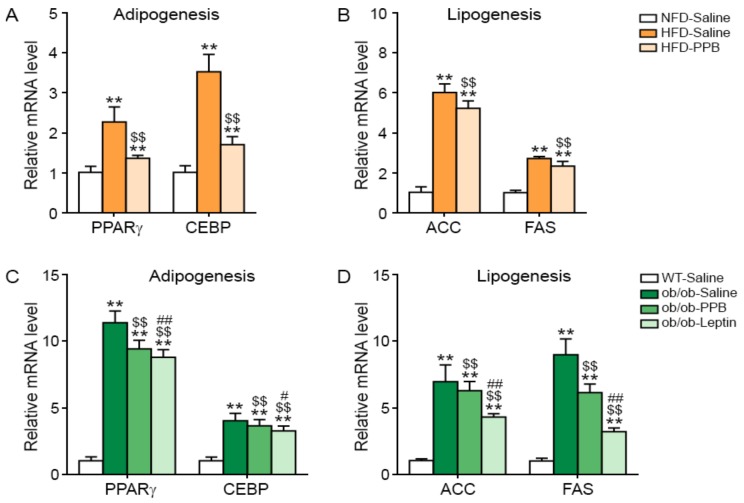
Reduction of adipogenesis and lipogenesis-related molecule expression in the brain of high fat diet-induced obese mice and ob/ob mice by PPB *from E. cava* mRNA levels of the adipogenesis-related molecules (**A**) PPARγ and CEBP, and lipogenesis-related molecules (**B**) ACC and FAS in the brain of high fat diet–induced obese mice were measured by qRT-PCR and (**C**) PPARγ and CEBP, and lipogenesis-related molecules (**D**) ACC and FAS in the brain of ob/ob mice were measured by qRT-PCR. All mRNA levels are expressed as relative levels normalized to β-actin of the NFD-Saline for high fat diet-induced obese mice group or WT-Saline for ob/ob mice. Significance represented as **, *p* < 0.01 versus NFD-Saline or WT-Saline; $$, *p* < 0.01 versus HFD-Saline or ob/ob-Saline; #, *p* < 0.05 and ##, *p* < 0.01 versus ob/ob-PPB. PPARγ, peroxisome proliferator-activated receptor gamma; CEBP, CCAAT enhancer–binding protein; ACC, acetyl-CoA carboxylase; FAS, fatty acid synthase; PPB, pyrogallol-phloroglucinol-6,6-bieckol.

**Figure 7 nutrients-11-02773-f007:**
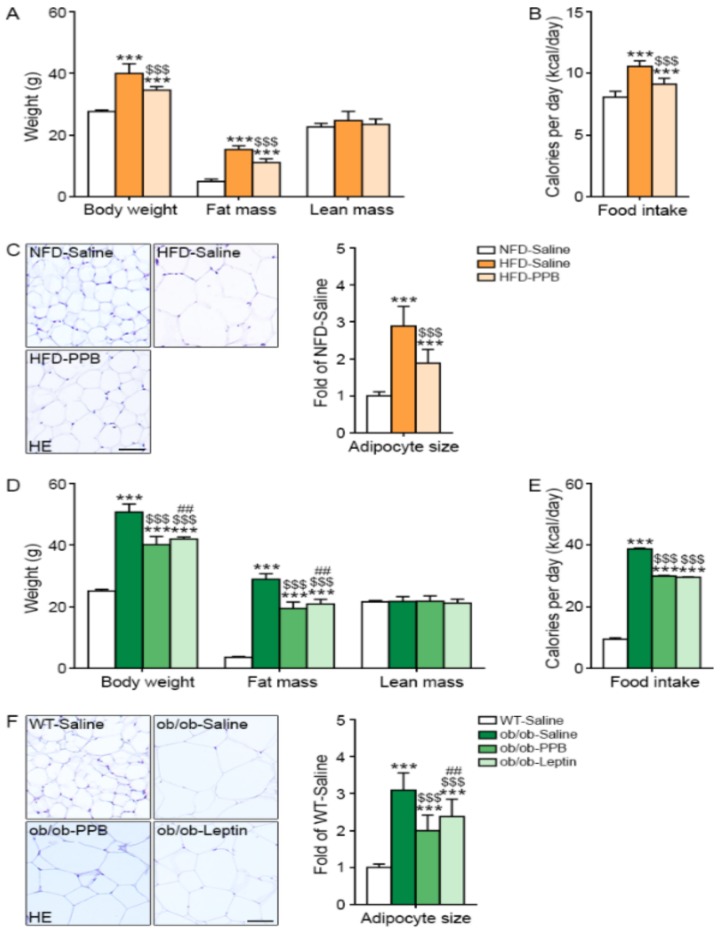
Reduction of body weight, food intake, and visceral fat size in high fat diet-induced obese mice and ob/ob mice by PPB from *E. cava* (**A**) Body weight, body composition (fat mass and lean mass), and (**B**) food intake of high fat diet–induced obese mice were measured before sacrifice. (**C**) Visceral fat was stained with hematoxylin and eosin (H&E). (**D**) Body weight, body composition (fat mass and lean mass), and (**E**) food intake of ob/ob mice were measured before sacrifice. (**F**) Visceral fat was stained with H&E. The size of the cells was quantified using Image J software and is shown as a graph. Scale bar = 50 µm. Significance represented as ***, *p* < 0.001 versus NFD-Saline or WT-Saline; $$$, *p* < 0.001 versus HFD-Saline or ob/ob-Saline; ##, *p* < 0.01 versus ob/ob-PPB. PPB, pyrogallol-phloroglucinol-6,6-bieckol.

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
