# Peer review of "Attenuation of Inflammation and Leptin Resistance by Pyrogallol-Phloroglucinol-6,6-Bieckol on in the Brain of Obese Animal Models"

_nutrients, 2019, doi:10.3390/nu11112773_

Round 1

Reviewer 1 Report

The authors of the submitted manuscript aimed to investigate the correlation between Pyrogallol-Phloroglucinol-6,6-Bieckol (PPB), a compound from the algae Ecklonia cava, and inflammation in the brain and adipose tissue. To evaluate this correlation, the authors compared a diet-induced obesity (DIO) animal model with a leptin-deficient (ob/ob) animal model. The authors also investigated whether the administration of PPB affected (active) macrophage infiltration, polarization of macrophages to M1, and pro-inflammatory cytokines in the brain and adipose tissue. Lastly, the authors aimed to examine whether changes in leptin resistance and/or leptin sensitivity occurred within the mice brains as a consequence of PPV administration.

The authors found that the administering PPB reduced inflammation, increased leptin sensitivity, and decreased leptin resistance, concluding that PPB may serve an important role in decreasing inflammation in both the brain and adipose tissue of mice by reducing the effects of the pro-inflammatory cytokines and recovering leptin sensitivity, which would thereby decrease food intake and mitigate weight gain.

Although the authors of the submitted manuscript should be commended for their work, this reviewer has some minor questions, comments, and points of clarification that need to be addressed before considering the manuscript further for publication consideration. To help guide and direct the author’s efforts, the reviewer has raised each question, point of clarification or concern point-by-point below. 

Introduction:

Line 20: Please consider defining what Ecklonia cava is in the introduction instead of line 67. Lines 45-46: “Molecular leptin sensitivity, defined as the ability of leptin to phosphorylate and 46 thereby activate STAT3 [3,6], is reduced after high-fat diet (HFD) intake”. Please consider moving this statement up one sentence to improve the flow of the paragraph for the reader. 

Materials and Methods:

Line 114: Dissimilar to the high-fat, diet-induced obese mice, the authors do not specify the leptin-deficient mice’s sex, or if there any notable differences between mice who were with different birth rates. The point about sex is particularly important as we recognize there are biological differences in how physiological responses and adaptations occur from sex differences.  

Other Comments:

This reviewer feels that the figures could be made clearer by sticking to a common color scheme throughout the entire manuscript, as opposed to having some figures in black and white and others in color. Additionally, this reviewer feels that the figure descriptions could be made clearer by stating what the figure is and why it is important to the study as a whole.  In Figure 5, it is stated that “Same letters represent no significant difference”. This seems to be unnecessary, as there are no variables labeled with letters within this figure.  The reviewer feels that having a list of all of the abbreviations on one page at the end of the manuscript would be helpful as a reference for readers.

Author Response

Point 1-1: Introduction: Line 20: Please consider defining what Ecklonia cava is in the introduction instead of line 67.

Response 1-1: We appreciated with this comment. As this comment, we rewrote defining what Ecklonia cava is in the introduction instead of line 67. Original sentence ‘Ecklonia cava (E. cava) is an edible marine brown alga, and is one of the richest natural sources of phlorotannins and phlorotannin derivatives that do not exist in land plants.’ was changed to ‘Ecklonia cava (E. cava) is a type of marine brown alga which is known to be one of the richest natural sources of phlorotannins.’ in Manuscript (line 67).

Point 1-2: Lines 45-46: “Molecular leptin sensitivity, defined as the ability of leptin to phosphorylate and 46 thereby activate STAT3 [3,6], is reduced after high-fat diet (HFD) intake”. Please consider moving this statement up one sentence to improve the flow of the paragraph for the reader.

Response 1-2: We appreciated with this comment. As this comment, “Molecular leptin sensitivity, defined as the ability of leptin to phosphorylate and 46 thereby activate STAT3 [3,6], is reduced after high-fat diet (HFD) intake.” move up in paragraph to improve the flow of the paragraph for the reader. This change can find in Introduction section of Manuscript (line 44-45).

Point 2: Materials and Methods: Line 114: Dissimilar to the high-fat, diet-induced obese mice, the authors do not specify the leptin-deficient mice’s sex, or if there any notable differences between mice who were with different birth rates. The point about sex is particularly important as we recognize there are biological differences in how physiological responses and adaptations occur from sex differences.

Response 2: We appreciated and agree with this comment. As this comment, authors did not specify the leptin-deficient mice’s sex in material and methods section. To confirm leptin treatment effects, researcher used male ob/ob mice for studying fertility capacity [Endocrinology. 1997 Mar;138(3):1190-3.], obesity [Endocrinology. 2017 Sep 1;158(9):2930-2943], bone loss [Sci Rep. 2019 Jun 27;9(1):9336.], lipodystrophy and insulin resistance [Endocrinology. 2018 Jun 1;159(6):2308-2323] etc. In this study, male six-week-old leptin-deficient obese (ob/ob) mice was used. We added that information in material and methods section (line 114).

Point 3-1: Other Comments: This reviewer feels that the figures could be made clearer by sticking to a common color scheme throughout the entire manuscript, as opposed to having some figures in black and white and others in color. Additionally, this reviewer feels that the figure descriptions could be made clearer by stating what the figure is and why it is important to the study as a whole. 

Response 3-1: We appreciated with this comment. To clearly understand to reviewer and reader, figure color (changing black & white to color) and method of statistical expression were changed.

1) In this study, we used two animal models. Orange color presented diet-induced obese (DIO) animal model  results and green color graph presented leptin deficiency mice (ob/ob) results. As well as graph color change, figure legends also were changed in manuscript (line 206, 216).

2) Used statistical expressions (for examples, a, b, c, d, etc) can confuse reviewer and reader. So these statistical expressions were changed and figure legends and material and methods section were rewrote. All markers indicated following; * (asterisk) means comparing with BSA, NFD-Saline or WT-Saline, $ comparing with PA-BSA, HFD-Saline or ob/ob-Saline, # comparing with PA-PPB or ob/ob-PPB (line 195, 206, 216).   

Point 3-2: In Figure 5, it is stated that “Same letters represent no significant difference”. This seems to be unnecessary, as there are no variables labeled with letters within this figure. 

Response 3-2: We appreciated with this comment. As this comment, “Same letters represent no significant difference” were removed and the change can find in figure legend 5 (line 240, 172).

Point 3-3: The reviewer feels that having a list of all of the abbreviations on one page at the end of the manuscript would be helpful as a reference for readers.

Response 3-3: We appreciated with this comment. As this comment, we added a list of all the abbreviations on one page at the end of the manuscript (line 333).

Reviewer 2 Report

In this manuscript, the authors treated macrophages and animals (DIO and OB/OB) with compound PPB and determined the effects of PPB on obesity induced inflammation, macrophage polarization, and leptin resistance in brain of obese animal models. This is an interesting paper if the authors can address the following questions.

1) Fig 1: What is the concentration/treatment time of PA-BSA? No details in the section of methods. TNFa mRNA signal is extremely high when compared to the value from our lab.

2) Fig 2: No group of "NFD-PPB" available. Macrophage polarization makers are derived from "fat", the contributions of adipocytes or ATMs (adipose tissue macrophages) are hard to tell. Fig 3 has the similar problems.

3) Section 2.3.1: The mice were fed HFD for 4 weeks, then PPB was administered for 4 weeks. The total time is 8 weeks based on the description of "after 8 weeks". The HFD effects can last for additional 4 weeks without further HFD feeding?

4) The authors presented data regarding macrophage activation. polarization, and cytokines level in adipose and brain tissue. However, there is no specific target tissue, no specific pathway, no interactions between macrophages and adipose and brain tissues. Similar, the authors showed TLR expression, ER stress markers, NF-kB expression, SOCS3 and STAT3 level, but the underlining mechanisms PPB works through are not determined. 

5) Minor languages problems with grammars.

Author Response

Point 1-1: Fig 1: What is the concentration/treatment time of PA-BSA? No details in the section of methods.

Response 1-1: We appreciated with this comment. In this study, 0.25 mM PA-BSA was treated to Raw 264.7 cells and 3T3L-1 cells for 48 hours. The information of concentration/treatment time of PA-BSA was added in Figure 1 legend and material and methods section. The changes can find in Manuscript (line 195, 91). 

Point 1-2: Fig 1: TNFa mRNA signal is extremely high when compared to the value from our lab.

Response 1-2: We appreciated with this comment. We think that even though the experiment is the same, the results of experiments can be different because different machines and reagents are used in different laboratories. However, (blow image A) we treated 0.25 mM PA-BSA to Raw 264.7 cell for 48 hours, and TNF-α increased about 12 times compared to BSA control. TNF-α mRNA levels in (blow image B) P388D1 (monocytes) (blow image C) Raw 264.7 cells increased about 14- or 16-fold compared to PBS control [Mar Drugs. 2018 Nov 9;16(11)]. In addition, the exact comparison is difficult, but the TNF-α level is also increased in PA treated Raw 264.7 cell compared to non-treated cell [PLoS One. 2014 Jul 21;9(7):e102373]. In addition, the results of this study were repeated three experiments to improve the reliability.

Point 2-1: Fig 2: No group of "NFD-PPB" available.

Response 2-1: We appreciated with this comment. We tried to find and correct the point you pointed out, but unfortunately we couldn't find it even though we thoroughly searched both the Figure 2 image and the manuscript. If you say that we did not find it would be cumbersome, but please point out again.

Point 2-2: Macrophage polarization makers are derived from "fat", the contributions of adipocytes or ATMs (adipose tissue macrophages) are hard to tell. Fig 3 has the similar problems.

Response 2-2: We appreciated with this comment. As your comment, macrophage polarization makers are derived from "fat", the contributions of adipocytes or ATMs (adipose tissue macrophages) are hard to tell. We can find out reference paper for adipose tissue macrophage markers in mouse and the reference shows TNF-α, IL-6, CD206 and CD11c etc as adipose tissue macrophage markers [Curr Opin Clin Nutr Metab Care. 2011 Jul; 14(4): 341–346.; J Clin Invest. 2007;117(1):175-184]. According to your comment, we additionally identified CD11c, which was not shown in our results. As you see in the image results below, The CD11c expression was increased by HFD (A, same with main figure 2 animal) and ob/ob mice (B, same with main figure 3 animal) and was significantly decreased by PPB in visceral fat. We added this result in supplementary figure 2 and described in result section of manuscript (line 196, 209).

Point 3: Section 2.3.1: The mice were fed HFD for 4 weeks, then PPB was administered for 4 weeks. The total time is 8 weeks based on the description of "after 8 weeks". The HFD effects can last for additional 4 weeks without further HFD feeding?

Response 3: We appreciated and agree with this comment. The mice were fed HFD for weeks first, and then PPB was administrated with HFD feeding for 4 weeks. So HFD feeding is totally 8 weeks. To make readers and reviewer clearer about our animal experiments, we created an experimental scheme image and added it to the supplementary file (Figure S1) and wrote legends and material and methods in manuscript (line 102).

Point 4: The authors presented data regarding macrophage activation. polarization, and cytokines level in adipose and brain tissue. However, there is no specific target tissue, no specific pathway, no interactions between macrophages and adipose and brain tissues. Similar, the authors showed TLR expression, ER stress markers, NF-kB expression, SOCS3 and STAT3 level, but the underlining mechanisms PPB works through are not determined.

Response 4: We are appreciate for your valuable comment. As this comment, we didn’t show you how to use the exact target tissue or pathway (fat and brain). However, this study showed changes in brain and adipose tissue because ER stress and inflammation are important in obesity and diabetes [Novartis Found Symp. 2007;286:86-94; Circ Res. 2010 Sep 3;107(5):579-91; Int J Obes (Lond). 2008 Dec;32 Suppl 7:S52-4]. Although PPB seems to work in both brain and fat, the fact that PPB does not directly work in the brain seems to be the limitation of this study. As a result showing that DK [Biomaterials. 2015 Aug;61:52-60.], one of the phlorotannins extracted from E. cava extract, acts directly in the brain beyond BBB, it is necessary to conduct similar studies in the future. We explained this point using proper reference and added this point in added in manuscript (line 328, 463).

Point 5. Minor languages problems with grammars.

Response 5: We appreciated this comment. As this comment, we checked thoroughly all language problems with grammars in manuscript. Some changes are marked with red color in manuscript.

Round 2

Reviewer 2 Report

First of all, this is nice work. But the reviewer has some concerns to talk about.

Macrophage polarization makers are derived from "fat", the contributions of adipocytes or ATMs (adipose tissue macrophages) are hard to tell. Fig 3 has similar problems.-------Sorry for that I did not describe my previous comments clearly. 

Fig 2 and 3: The authors investigated the effects of PPB on macrophages (infiltration and polarization), but the materials are fat and brain tissues. The detected markers are the ones from macrophages. These tissues contain local macrophages and local adipocytes/brain cells. So it is hard to tell how much contributions to the signals are from local macrophages and how much from local adipocytes/brain cells. The best thing is to isolate local macrophages (for example ATM) and detect those markers.

Another question is the target of PPB. What is the specific target? Macrophages? Fat cells? Brain cells? PPB targets macrophages and then affects adipocytes/brain cells? or vice versa??This is the first step to the underlying mechanisms. Maybe PPB targets everywhere (we don't know)...but a focus is necessary for a manuscript...

Author Response

Point 1: Fig 2 and 3: The authors investigated the effects of PPB on macrophages (infiltration and polarization), but the materials are fat and brain tissues. The detected markers are the ones from macrophages. These tissues contain local macrophages and local adipocytes/brain cells. So it is hard to tell how much contributions to the signals are from local macrophages and how much from local adipocytes/brain cells. The best thing is to isolate local macrophages (for example ATM) and detect those markers.

Response 1: Thank you very much for your meticulous comment. As your comment, the materials are fat and brain tissues and these tissues contain local macrophages and local adipocytes/brain cells. In this study, we investigated the effects of leptin resistance and leptin sensitivity on two animal models (diet induced obese mice model and leptin deficiency mice model) using PPB administration. Orally administered PPB reduced macrophage activation and polarization in visceral fat and decreased inflammatory cytokine expression. Inflammation in visceral fat relieved TLR4 expression and endoplasmic reticulum (ER) stress in brain tissues, thereby reducing leptin resistance and leptin sensitivity such as NF-κB, SOCS3, pSTAT3, and Ob-R. Eventually, brain tissue changes confirmed adipogenesis and lipogenesis in the visceral fat and body weight gain, fat mass, food intake, and visceral fat size. As mentioned above, it was important to confirm the increase of inflammatory macrophages and secretion of inflammatory cytokines in visceral adipose tissue. Distinguishing local macrophages or local adipocytes / brain cells, as your comment suggests, will be actively taken into account in the future study. This content will be written in the discussion section to address the limitations of future research and this research. The changes can find in Manuscript (line 270).

Point 2: Another question is the target of PPB. What is the specific target? Macrophages? Fat cells? Brain cells? PPB targets macrophages and then affects adipocytes/brain cells? or vice versa??This is the first step to the underlying mechanisms. Maybe PPB targets everywhere (we don't know)...but a focus is necessary for a manuscript...

Response 2: We are appreciated for your valuable comment and agreed with comment. We confirmed that oral administration of PPB alleviates leptin resistance and sensitivity in the brain through changes in inflammatory macrophages and cytokines in adipose tissue. So far, research has shown that Dieckol, one of the phlorotannins of Ecklonia cava extract, passes directly through the brain-blood barrier of the brain and acts directly on the brain [Biomaterials. 2015 Aug;61:52-60.]. Therefore, the PPB used in this study could be predicted to be possible, but unfortunately it was not shown in this study. Therefore, the authors will further study to identify the exact target pathway or target tissue of PPB in future studies. We explained this limitation using proper reference and added this point in added in manuscript (line 326).
